# Investigation of Structural, Morphological, and Optical Properties of Novel Electrospun Mg-Doped TiO_2_ Nanofibers as an Electron Transport Material for Perovskite Solar Cells

**DOI:** 10.3390/nano13152255

**Published:** 2023-08-05

**Authors:** Kubra Erdogar, Ozgun Yucel, Muhammed Enes Oruc

**Affiliations:** 1Department of Physics, Gebze Technical University, Gebze 41400, Turkey; kerdogar@gtu.edu.tr; 2Department of Chemical Engineering, Gebze Technical University, Gebze 41400, Turkey

**Keywords:** nanofibers, Mg-doped TiO_2_, perovskite, electron transport material

## Abstract

Perovskite solar cells (PSCs) are quickly becoming efficient solar cells due to the effective physicochemical properties of the absorber layer. This layer should ideally be placed between a stable hole transport material (HTM) layer and a conductive electron transport material (ETM) layer. These outer layers play a critical role in the current densities and cell voltages of solar cells. In this work, we successfully fabricated Mg-doped TiO_2_ nanofibers as ETM layers via electrospinning. This study systematically investigates the morphological and optical features of Mg-doped nanofibers as mesoporous ETM layers. The existence of the Mg element in the lattice was confirmed by XRD and XPS. These optical characterizations indicated that Mg doping widened the energy band gap and shifted the edge of the conduction band minimum upward, which enhanced the open circuit voltage (*V_oc_*) and short current density (*J_sc_*). The electron-hole recombination rate was lowered, and separation efficiency increased with Mg doping. The results have demonstrated the possibility of improving the efficiency of PSCs with the use of Mg-doped nanofibers as an ETM layer.

## 1. Introduction

The broad use of technology and the increase in world population have caused extensive energy demands and environmental awareness in developed and developing countries. Solar energy has a significant capability for future energy that can aid in reducing fossil fuel use and growing environmental concerns such as global warming [1]. In recent years, perovskite solar cells (PSCs) have attracted intensive attention because of their ever-increasing power conversion efficiency (PCE), low-cost material constituents, and simple solution fabrication processes [2,3]. The conventional configuration of perovskite solar cells has five layers: (i) transparent conducting oxide (TCO), which can be made of fluorine-doped tin oxide (FTO) or indium tin oxide (ITO), (ii) the planar or mesoporous electron transporting material (ETM), (iii) the light-absorbing material made of perovskite materials, (iv) the hole transporting material (HTM), and (v) the electrode (Figure 1a). The perovskite layer absorbs the incident of light, generating electron-hole pairs, and subsequently, the electrons are transported to the ETM layer, while the holes are transported to the HTM layer. These charge carriers are finally collected by electrodes forming a photovoltage, which depends on the gap between the conduction band minimum (CBM) of ETM and the valence band maximum (VBM) of HTM (Figure 1b) [4]. Therefore, tuning the energy band gaps and the band edge positions of ETM and HTM layers is of critical importance to the performance of PSCs.

The ETM layer has two main roles in the electron transport mechanism; the first role is to receive the photoexcited electrons from the perovskite absorber and transport the collected charges to the conductive layer, and the second role is to block the generated holes [4]. Titanium dioxide (TiO_2_) is widely used as an ETM due to its favorable band gap positioning, stability, and low cost in perovskite solar cells [5]. To replace TiO_2_ as an ETL layer in PSCs, scientists and researchers are investigating other types of inorganic materials, such as metal oxides like ZnO, SnO_2_, WO_3_, In_2_O_3_, CuO, and Cu_2_O [6,7]. ZnO, which is one of the best candidates due to a number of reasons, is not the optimal material to replace TiO_2_ as the ETL in PSC devices. For example, ZnO is naturally hygroscopic, which could lead to perovskite disintegration in ambient air [8]. PSCs based on low-temperature processed ZnO as an ETL were additionally environmentally fragile because any hydroxyl residues that remained on the ETL surface could hasten the breakdown of the perovskite structure [9].

Two types of structures can be used in the architecture of an ETM layer; planar (bl-TiO_2_) or mesoporous (mp-TiO_2_) layers [6]. In a planar structure, TiO_2_ is deposited on FTO film as a compact layer. With the implementation of this geometry, Granados et al. obtained 6.29% of PCE for an FTO/bl-TiO_2_/mp-TiO_2_/Perovskite/P3HT/Au cell structure [10]. However, it is crucial to fabricate a mesoporous structure to achieve high-efficiency perovskite solar cells. Having a larger surface area offers the benefit of facilitating a more significant interaction between the ETM and the perovskite active layer, thereby promoting faster charge transfer kinetics. In modern mesoporous perovskite solar cells (mp-PSCs), TiO_2_ nanoparticles are often employed as nanostructured scaffolds. Because of their labyrinthine shape and the presence of grain boundaries that interfere with the electron transport channels within mp-TiO_2_ nanoparticles, the mesoporous layer that uses nanoparticles might have difficulty in loading perovskite [11]. To solve this problem, numerous one-dimensional TiO_2_ nanostructures have been used as ETLs in PSCs, including nanorods, nanowires, nanotubes, and nanocones [12]. Among the mesoporous fabrication methods, electrospinning is a simple, inexpensive technique by which an ETM layer can be fabricated [13]. Dharani et al. used electrospun TiO_2_ fibers as an ETM layer leading to a device efficiency as high as 9.8% [14].

As an ETM layer, TiO_2_ doped with metal ions such as Mg, Li, Y, and Nb have been utilized to increase the efficiency of the system to over 19% as a result of improving the extraction and/or injection of carriers [15,16,17,18]. Amalathas et al. studied Li-doped mesoporous (mp-TiO_2_) and obtained a 17.59% power conversion efficiency (PCE) [15]. Roose and colleagues studied mesoporous (mp-TiO_2_) Nd-doped electrodes for perovskite-based solar cells and obtained an 18.2% PCE [19]. A PCE of 19.8% was achieved with Mg-doped TiO_2_ electrodes in perovskite solar cells [20]. The Mg-doped TiO_2_, which had a larger band gap and lower resistance compared to the non-doped TiO_2_, has important advantages [21]. Firstly, a higher band gap showed better optical transmission properties, allowing a higher number of photons to excite the perovskite material leading to elevating the short current density (*J_sc_*) [22]. Secondly, Mg doping lifted up the CBM and deflated the VBM of the ETM layer. This enlargement provided PSCs with higher values of open circuit voltage (*V_oc_*) and fill factor (FF) [23]. Third, the shift in CBM provided a better energy-level alignment at the interface, which could be quantitatively stated by the conduction band offset (CBO): the difference between the energy levels of the CBM of the ETL and that of the perovskite layer [24]. This alignment suppressed the electron-hole recombination and improved both the *V_oc_* and *J_sc_* when employed in PSCs. 

Mg-doped TiO_2_ nanofibers as an ETM layer fabricated by the electrospinning method have not been investigated yet. There are few reports available for Mg-doped TiO_2_ nanoparticles to be carried out through a complex sol–gel process, which is expensive and slow and was deposited using the spin coating method [20,21]. 

To the best of our knowledge, this is the first study that successfully shows the fabrication of Mg-doped TiO_2_ nanofibers that can provide a larger interfacial contact area to support film deposition as a mesoporous ETM layer. In addition, a more homogeneous structure was obtained by doping TiO_2_ nanofibers with Mg ions before the electrospinning process. The effect of the calcination temperature and time on the crystal structure, morphology, and optical properties of the nanofibers was also investigated. After the calcination of fibers at 500 °C, XRD analyses were carried out to reveal the crystal structure and UV-Vis absorption was utilized to determine the band gap of each sample.

## 2. Materials and Methods

### 2.1. Materials

Titanium tetraisopropoxide (TTIP), polyvinylpyrrolidone (PVP; Mw 1,300,000), ethanol, and acetic acid were obtained from Sigma Aldrich. Magnesium chloride (MgCl_2_) was obtained from Merck. TTIP and MgCl_2_ were used as precursors. Ethanol and distilled water were used as a solvent. FTO glass (10 Ω per square) with the dimensions 25 mm × 75 mm × 2.2 mm was obtained from Teknoma Technological Materials Industrial and Trading Inc.

### 2.2. Experimental

In the fabrication of the ETM layer, firstly, FTO substrates were cleaned ultrasonically in acetone, isopropyl alcohol, and ethanol for 15 min before being sequentially and finally dried under nitrogen medium (99.9% purity). The substrates were exposed to air plasma (at 300 mTorr pressure) for 15 min just before the deposition of the compact layer. For the compact layer, the precursor solution was prepared by mixing two solutions. The first solution was composed of 1.5 mL TTIP and 4.5 mL ethanol (≥99.9%), stirred at 1000 rpm for 15 min. The second solution was composed of 1 mL acetic acid (99.8–100.5%) and 3 mL ethanol, stirred at 1000 rpm for 15 min. The second solution was added to the first solution drop by drop and stirred for 30 min at 1000 rpm. Then, the mixed solution aged for one day. During aging, the solution was covered with aluminum foil and stored in a dark medium. Before the deposition step, the solution was diluted by absolute ethanol 1:1 *v*/*v*. The compact layer was deposited two times on FTO glass by spin coating for 30 s at 3000 rpm and was annealed at 120 °C for 10 min. After the preparation of the compact layer, which provided better adhesion of the fibers when formed on the surface, the electrospinning process was carried out.

Figure 2 shows the schematic illustration of the TiO_2_ nanofiber fabrication process. Step I shows the polymer solution preparation. For non-doped TiO_2_ nanofibers, 0.25 g of titanium tetraisopropoxide was hydrolyzed with a mixture of 1.4 mL ethanol and 0.6 mL acetic acid and aged for 48 h. The PVP (9 wt.%) was dissolved in 2.75 mL ethanol for three hours. Then, the TiO_2_ solution was added drop by drop to the PVP solution and stirred for half an hour. For the Mg-doped TiO_2_ fibers, PVP was dissolved in 2.75 mL of ethanol, and then MgCl_2_ was added to the PVP solution at various concentrations (0.1% and 0.5%). Then, the TiO_2_ solution, as prepared before, was added drop by drop to the MgCl_2_-PVP solution. The mixed precursor solution was stirred for half an hour at room temperature to attain the sufficient viscosity required for electrospinning.

For the electrospinning process, the solutions were loaded into a syringe with a 25-gauge (25 G) needle, and the distance between the needle and collector was adjusted to 10–15 cm. A direct current was applied at 10–20 kV, and the feed rate of the precursor solutions was adjusted to 0.5 mL/h, controlled by using a syringe pump. The humidity of the air was around 50%. Aluminum foil was used for nanofiber collection and for optical and microscopic measurements. Fiber diameters were measured from SEM images using the ImageJ analysis program. Finally, all fibers were calcined at 500 °C [25,26] for 2 h in a Protherm PTF 12/40/250 tube furnace for the transition from the amorphous structure to the crystal structure.

## 3. Results and Discussion

### 3.1. SEM Characterization

The surface morphology of TiO_2_-PVP nanofiber samples before calcination was investigated to optimize the electrospinning operating conditions. Figure 3 shows SEM images of electrospun TiO_2_-PVP nanofibers when fabricated by a solution of ethanol/acetic acid/PVP/Ti(OiPr)_4_ at a feed rate of 0.5 mL/h. Uniform fibers with different diameters were achieved at various voltage fields and collector distances. The fiber diameters decreased with an increase in the applied electric field and increased with an extension in the collector distance. The obtained diameters are consistent with the literature [27].

The solutions of ethanol/acetic acid/PVP/MgCl_2_/Ti(OiPr)_4_ including 0.1% (wt.) MgCl_2_ and 0.5% (wt.) MgCl_2_ were used to produce Mg-doped TiO_2_ fibers. It is known that after the calcination process, a shrinkage in the nanofiber diameter can be observed [14]; therefore, fabricating higher diameter nanofibers compared to TiO_2_-PVP nanofibers was crucial. After our preliminary trials, the nanofibers were deposited on a glass substrate at a potential of 20 kV, a distance of 15 cm, and a feed rate of 0.5 mL/h. Then, nanofibers were calcinated at 500 °C for 2 h on the glass substrate for SEM characterization and in a crucible for XRD measurements.

Figure 4 depicts the SEM images of the TiO_2_-PVP nanofibers with MgCl_2_ and Mg-doped TiO_2_ nanofibers after calcination. Table 1 summarizes the nanofiber’s diameters and shrinkage percentages. The mean fiber diameters for samples A and C without calcination were 249 ± 68 nm and 262 ± 88 nm, and the sample area was 2365 nm^2^ and 972 nm^2^, respectively. The higher amount of Mg-doped in sample C led to a slight increase in the fiber diameter. On the other hand, after the calcination process for samples B and D, the fibers shrank, and a significant decrease in the diameter was observed. The diameters of samples B and D were measured as 80 ± 30 nm and 134 ± 32 nm, and the sample area 306 nm^2^ and 499 nm^2^, respectively. The shrinkage percentages of samples with 0.1% and 0.5% MgCl_2_ were 68% and 50%, respectively. However, it was concluded that a higher amount of Mg restricted the shrinkage percentages. This effect could be attributed to the insertion of the dopants into the TiO_2_ lattice, which might cause lattice and volume expansion [28].

The EDX elemental mapping used to determine the homogeneous distribution of the Mg element is shown in Figure 5. Before and after the calcination process, Mg element mapping showed a uniform distribution in the nanofibers. These images indicate that Mg doping was achieved throughout the nanofibers. The cross-section view of the nanofibers after calcination can be seen in Figure 5c.

### 3.2. XRD and XPS Analysis

TiO_2_ is a polymorphous material and has three crystalline phases, namely anatase, brookite, and rutile [29]. A stronger surface interaction and simpler charge transfer were also indicated by the fact that the bond lengths in a rutile TiO_2_-perovskite system were shorter than those in an anatase TiO_2_-perovskite system [30]. XRD techniques were used to examine the crystal structure and phase evolution of the electrospun TiO_2_ nanofibers (Rigaku Slab Cu k-alpha 1.58 A Xray). For the phase evolution, Figure 6a,b show the XRD patterns for non-doped TiO_2_ nanofibers for different calcination times, and Figure 6c,d show 0.1% and 0.5% Mg-doped nanofibers, respectively. Mg and MgO peaks are indicated [31]. The as-spun nanofibers were amorphous. Well-crystallized anatase-rutile TiO_2_ and Mg-doped TiO_2_ nanofibers were obtained by heating the as-spun fibers at 500 °C. It was observed that at 2 h calcination time, the rutile phase occurred more clearly. In Figure 6c,d, the Mg peak was observed at 44° (012), as consistent with the literature [32]. However, the doping of Mg drove the direction of rutile to anatase as expected, and the rutile phase intensity was reduced dramatically. Because the Mg^2+^ ions were incorporated in the anatase lattice, this resulted in Mg-doped TiO_2_ nanofibers with a higher conduction band.

We could determine the crystal size (D) using the Scherer relation D = k × λ/β × (cosθ), where k is the constant (0.9), the X-ray wavelength (0.154 nm) is β, and the FWHM value is the reflection angle. The crystal size of undoped TiO_2_ and MgCl_2_-doped TiO_2_ films is displayed in Table 2.

As shown in Table 2, when the amount of doping increased, there was a small reduction in the crystallite size of the crystal structure. This was because some of the Mg dopant ions did not proceed into the TiO_2_ lattice and instead remained on the surface, creating grain boundaries. This could impact the growth of the crystal structure, resulting in a smaller crystal size as the doping concentration increased. However, if the amount of doping increased too much, it could inhibit crystal growth and increase the number of grain boundaries. This could have a negative impact on the performance of the material as a photo-anode [33]. In addition, the smaller crystal sizes reduced these defects and generally resulted in a larger surface area and higher porosity, which could improve the contact between the ETL and the perovskite layer, leading to improved device performance.

In Figure 7, there is an increase in the binding energy (BE) in Mg-doped TiO_2_ relative to non-doped TiO_2_. For the XPS fittings, adj.R-squared values ranged between 0.95 and 0.98. This shift showed the interaction of the Ti element with the dope material. The charge transfer through Mg doping could result in a slight shift of BE without the formation of different phases (MgO) [22]. The Ti 2p1/2 and Ti 2p3/2 binding energy of Mg-doped TiO_2_ was higher than that of non-doped TiO_2_, and this meant that due to the stronger metallicity and lower valence state, maybe Ti^4+^ could donate electrons easier than Mg^2+^. For MgO, the 458.7 eV curve shows 3/2 Ti^0^, and 460.7 eV shows ½ Ti^0^ (Figure 7e,f).

### 3.3. Optical Band Gap and UPS Characterizations

The optical properties of non-doped TiO_2_ and Mg-doped TiO_2_ nanofibers were characterized by a UV-Vis spectrophotometer. The optical responses of these samples are displayed in Figure 8a. The compact layer was used as a base measurement. The results show that samples with calcinated nanofibers had a higher absorbance in UV and visible regions. This resulted from the interaction of light with a higher surface area of nanofibers. Among samples with nanofibers, the absorbance of Mg-doped TiO_2_ nanofibers was less than that of non-doped TiO_2_ nanofibers from 320 to 900 nm. On the other hand, for the wavelength below 320 nm, the opposite behavior was observed. This absorption capability is reported in the literature [34]. It is a result of Mg doping that shifts the absorption edge from a visible to a UV region. It can be stated that the substitution of Mg in Ti sites and the formation of Mg-Ti mixed oxides could enlarge the band gap [33]. Therefore, Mg doping enhances the absorption of light by the perovskite layer with the improved *J_sc_* when the incident light passes through the ETM layer. The absorption onset appeared to be blue-shifted with the increasing concentration of Mg. However, the light absorption of 0.5% MgCl_2_ doped films deteriorated, which could be attributed to the crystal size.

The optical band gaps were derived from the Tauc plot [35]. The Tauc relation is $\left(ahv^{1/2}\right)=A\left(hv-E_{g}\right)$, where a is the absorption coefficient, A is the absorption constant, $E_{g}$ is the band gap and hv is the photon energy; h is Planck’s constant, v is the frequency of light, and ½ is for the indirect band gap energy. The measured band gaps of non-doped TiO_2_, 0.1% Mg-doped TiO_2_, and 0.5% Mg-doped TiO_2_ nanofiber films are presented in Figure 8b. The non-doped TiO_2_ nanofiber’s band gap was 3.20 eV, and Mg-TiO_2_ showed a band gap value of 3.27 eV, which is consistent with the value reported in the literature [36]. When the amount of Mg doping was increased to 0.5%, a small increase in the band gap from 3.27 eV to 3.28 eV was obtained.

The changes in absorption and band gap tuning that occurred could be linked to the creation of trapping states and intermediate sources that can transport electrons. The addition of Mg as a dopant resulted in the formation of oxygen vacancies and localized trap states, which helped transport electrons. Therefore, the vacancies generated by these dopant atoms were highly effective for adjusting the material’s optoelectronic properties. The introduction of Mg as a dopant changes the structure of the material by rearranging the atoms into their respective lattice positions. As a result, the low bandgap energy of the Ti nanofibers is likely due to the larger average crystal size. When Mg is added to TiO_2_ nanofiber films, the crystal size decreases, and the band gap shifts. This increment in the band gap between the conduction and valence bands was anticipated to improve the injection of charges from the perovskite layer to the ETL.

In order to achieve the relatively accurate position of energy bands, UPS characterization was performed. Figure 9a depicts the UPS spectra obtained using He I radiation. The VBM value is calculated by the sum of the work function and the peak value at the extended valance spectra. The work function is derived from subtracting the cut-off banding energy (the right-hand side of the spectra) from the photon energy (21.21 eV). The work function of non-doped TiO_2_ can be calculated as 5.19 eV, that of doped TiO_2_ with 0.1% Mg as 5.05 eV, and that of doped TiO_2_ with 0.5% Mg as 5.21 eV. The extended valance spectra are displayed on the left-hand side of the spectra. The non-doped TiO_2_ film, the 0.1% Mg-doped TiO_2_ film, and the 0.5% Mg-doped TiO_2_ film, respectively, had peaks centered at 1.52, 1.67, and 1.51 eV. The VBM value was calculated by the sum of the work function and the peak value at the extended valance spectra. Therefore, the VBM positions of non-doped TiO_2_, doped TiO_2_ with 0.1% Mg, and doped TiO_2_ with 0.5% Mg were 6.71 eV, 6.72 eV, and 6.72 eV, respectively. The wider band gap that occurred in Mg-doped TiO_2_ nanofibers was due to an ascending shift in CBM [22].

### 3.4. Photoluminescence (PL) Spectra

Photoluminescence (PL) is the spontaneous emission of light from a material following optical excitation. It is a powerful technique to investigate the recombination rate of photoexcited electron-hole pairs in semiconductor materials. The PL spectrum is a result of the irradiative recombination of the photoexcited electron and hole; thus, a lower PL intensity indicates a lower electron-hole recombination rate. Figure 10 indicates that the intensity of Mg-doped TiO_2_ nanofiber samples was lower than those of non-doped TiO_2_. This behavior implies that for the Mg-doped sample, the electron-hole recombination rate was much lower, and separation efficiency was much higher.

The suppression of electron-hole recombination is critical for the energy-level alignment at the interface of TiO_2_ and perovskite layers. When the CBM of the TiO_2_ is lower than that of the perovskite (cliff structure), interface recombination becomes dominant, and the *V_oc_* declines. When the CBM of the TiO_2_ is higher than that of the perovskite (spike structure), the interfacial recombination is largely cut off, and an increase in the *V_oc_* of the solar cells is achieved [37]. With Mg doping, the cliff structure is minimized, which suppresses the electron-hole recombination and increases the efficiency of the perovskite cells. Figure 9b shows that the CBM of Mg-doped TiO_2_ samples increased with diminishing the cliff structure. Therefore, the decrease in the PL intensity was a result of this CBM tuning.

## 4. Conclusions

In this work, TiO_2_ nanofibers with different Mg doping concentrations were successfully fabricated using the electrospinning technique for the first time as the ETM layer. The nanofibers became crystalline after calcination at 500 °C for 2 h. From XRD patterns, anatase and rutile phases of TiO_2_ nanofibers were obtained. According to SEM analysis, nanofiber diameters lessened during calcination. XRD results confirmed Mg doping in anatase TiO_2_ with a structural change in terms of increasing crystallinity due to Mg doping. The presence of the Mg dopant in the TiO_2_ was also confirmed by the shift in Ti2p peaks in the XPS spectrum. The nanofiber samples, compared with the compact layer samples, showed a higher absorbance in UV and visible regions due to the interaction of light with an enhanced surface area. The absorption coefficient of Mg-doped TiO_2_ nanofibers became less than that of non-doped TiO_2_ nanofibers in the visible spectrum. Mg doping shifted the absorption edge from the visible to the UV region. Therefore, when incident light passed through the ETM layer, this shift increased the perovskite layer’s capacity to absorb light with an improved J_sc_. Adding Mg into the ETM layer not only increased the energy band gap, which contributed to the improvement of the *V_oc_* but also elevated the position of the CBM of TiO_2,_ which lowered the electron-hole recombination rate and high *J_sc_* based on PL measurements. The TiO_2_ mesoporous layer’s porosity is a crucial component in the development of perovskite crystal, which is crucial for charge carrier extraction and light harvesting effectiveness. The TiO_2_ mesoporous layer’s porosity was obtained at around 50%. In future work, we aim to fabricate the perovskite solar cell to investigate the performance of the cell in which perovskite infiltrates through the Mg-doped TiO_2_ ETM layer.

## Figures and Tables

**Figure 1 nanomaterials-13-02255-f001:**
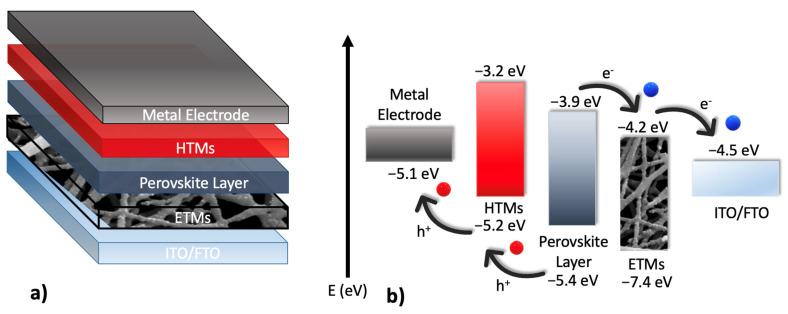
(**a**) The general configuration and (**b**) Energy level diagram of Perovskite solar cells.

**Figure 2 nanomaterials-13-02255-f002:**
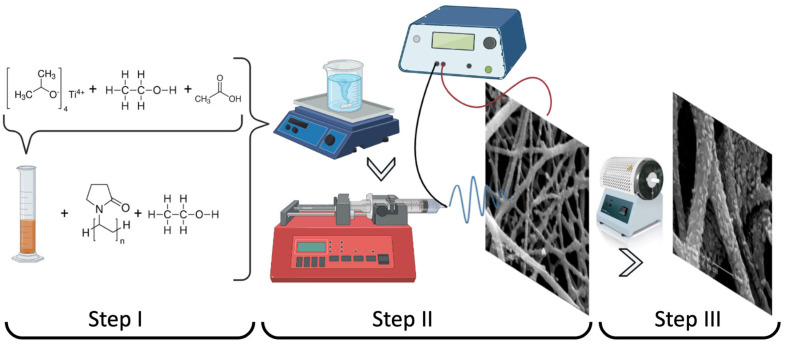
Fabrication steps of electrospun TiO_2_ nanofibers: (**I**) Polymer solution preparation, (**II**) Electrospinning process and (**III**) Calcination process.

**Figure 3 nanomaterials-13-02255-f003:**
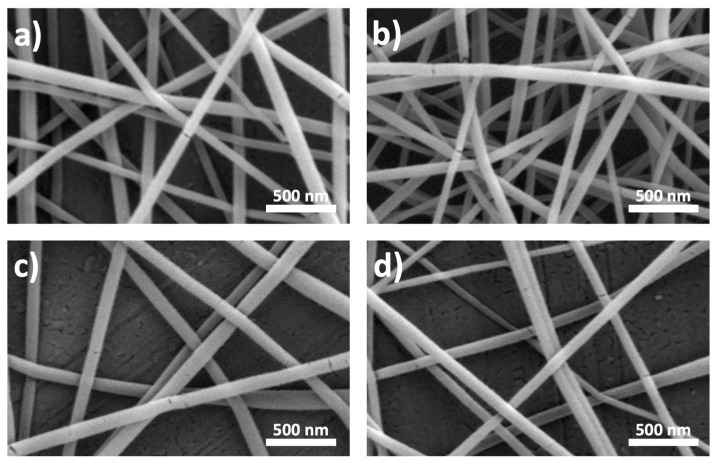
SEM images of the electrospun TiO_2_-PVP nanofibers (**a**) Voltage: 10 kV, Distance: 10 cm, Diameter: 85 nm (**b**) Voltage: 20 kV, Distance: 10 cm, Diameter: 70 nm (**c**) Voltage: 10 kV, Distance: 15 cm, Diameter: 108 nm (**d**) Voltage: 20 kV, Distance: 15 cm, Diameter: 74 nm.

**Figure 4 nanomaterials-13-02255-f004:**
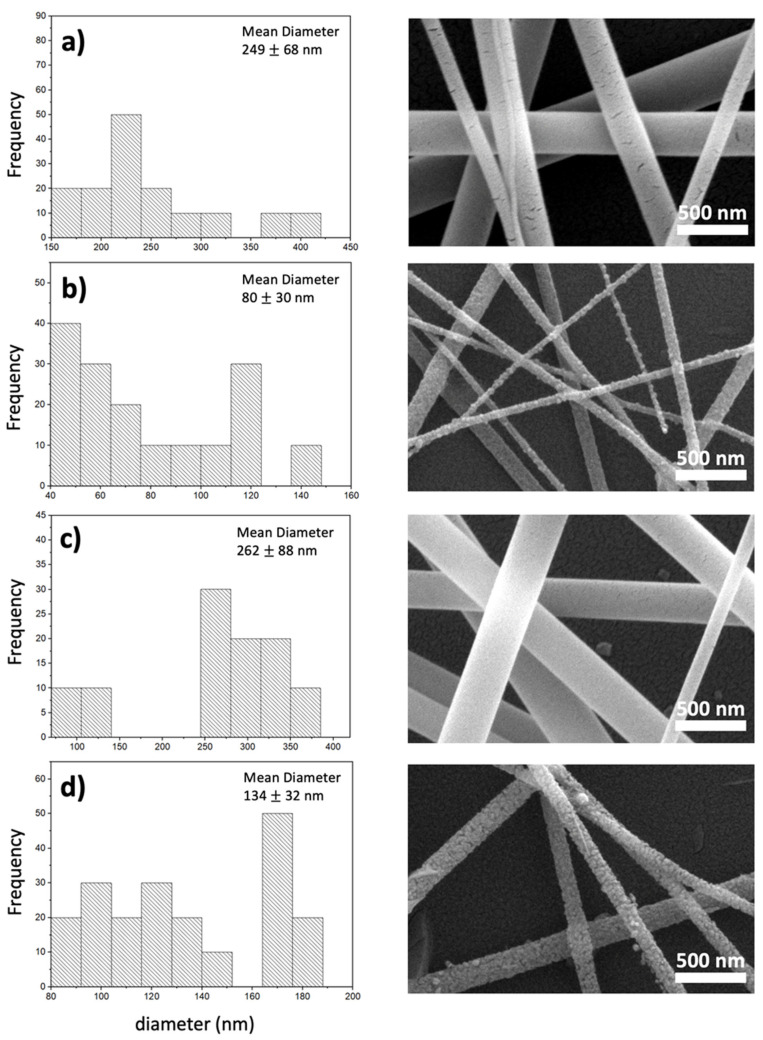
SEM micrographs of the electrospun nanofibers (**a**) TiO_2_-MgO (0.1%) before calcination (**b**) TiO_2_-MgO (0.1%) after 500 °C calcination, (**c**) TiO_2_-MgO (0.5%) before calcination, (**d**) TiO_2_-MgO (0.5%) after 500 °C calcination.

**Figure 5 nanomaterials-13-02255-f005:**
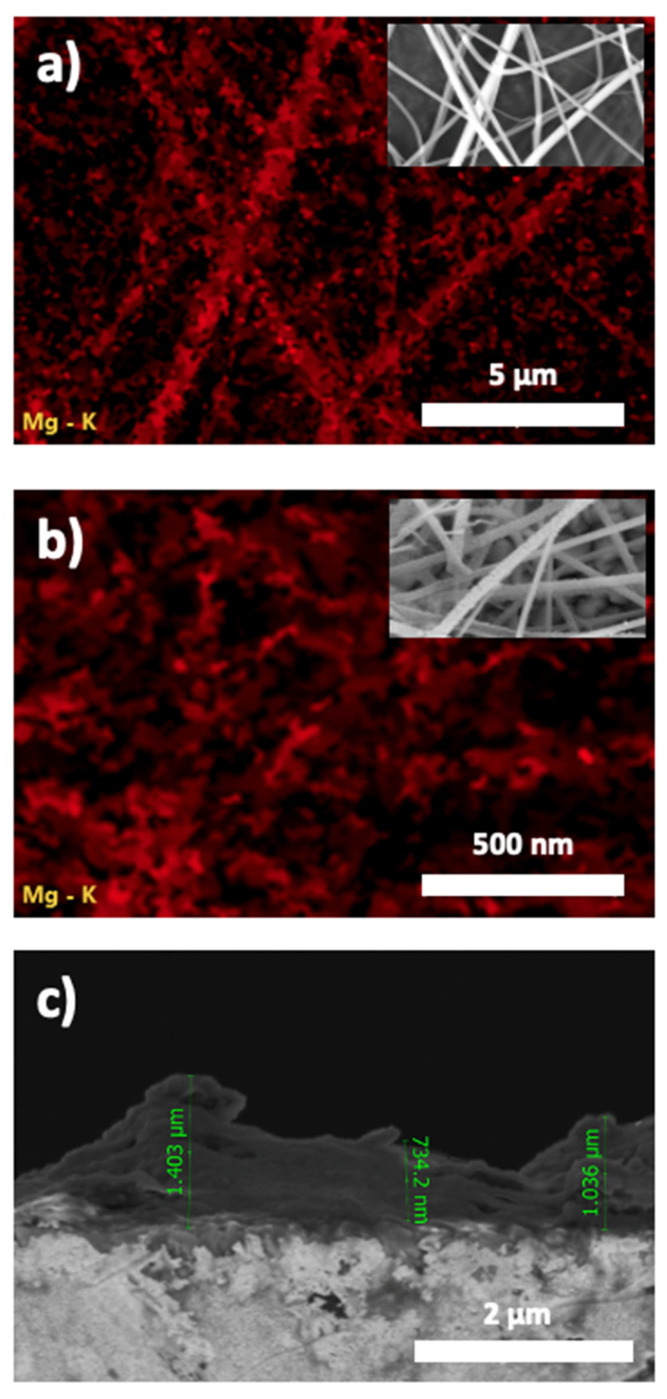
Images of SEM/EDX elemental mapping of Mg-doped fibers (**a**) Before calcination and (**b**) After calcination; the insets are the corresponding field emission SEM images. (**c**) Cross section of the nanofibers after calcination.

**Figure 6 nanomaterials-13-02255-f006:**
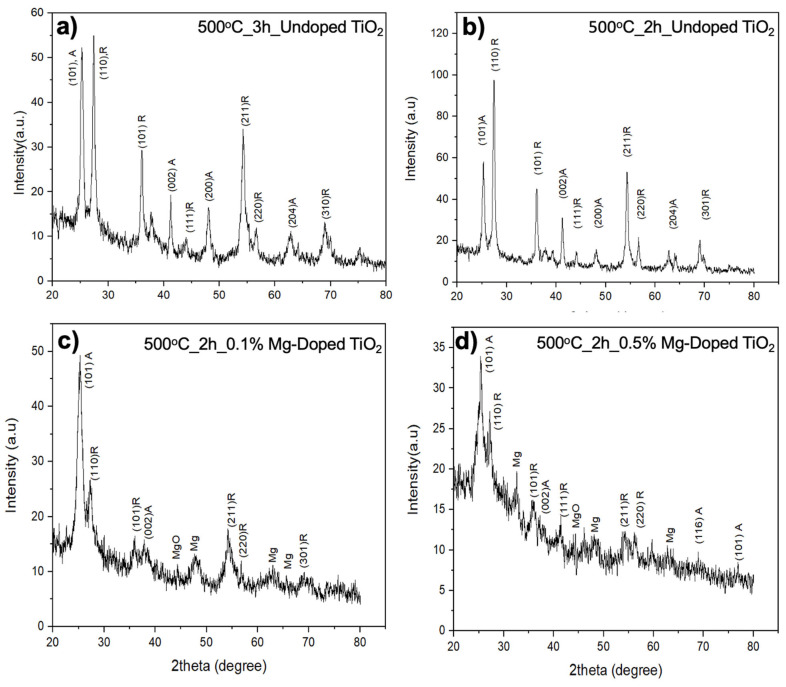
XRD patterns show the evolution of the anatase (101 A, 002 A, 200 A, 204 A, 116 A, 101 A) and rutile (110 R, 101 R, 111 R, 211 R, 220 R, 301 R) for (**a**–**d**) and (012) Mg for (**c**,**d**).

**Figure 7 nanomaterials-13-02255-f007:**
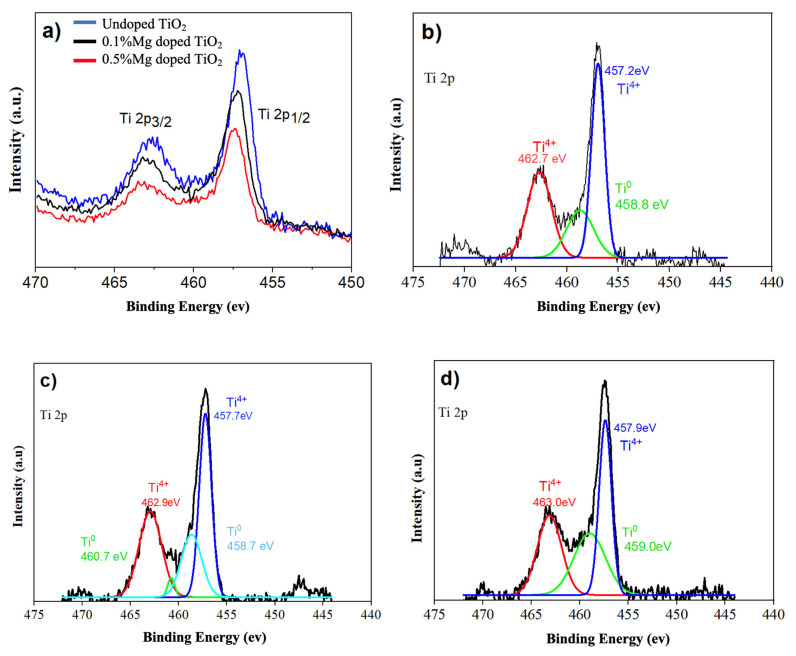
XPS analyses of the Ti 2p for (**a**) non-doped TiO_2_, 0.1% Mg-doped TiO_2_, and 0.5% Mg-doped TiO_2_ nanofibers. (**b**) Non-doped TiO_2_ fitting for (**c**) 0.1% Mg-doped TiO_2_ and (**d**) 0.5% Mg-doped TiO_2_. (**e**) Mg 1s binding energy for 0.1% Mg doped TiO_2_ and (**f**) Mg 2p MgO binding energy for 0.1% Mg doped TiO_2_.

**Figure 8 nanomaterials-13-02255-f008:**
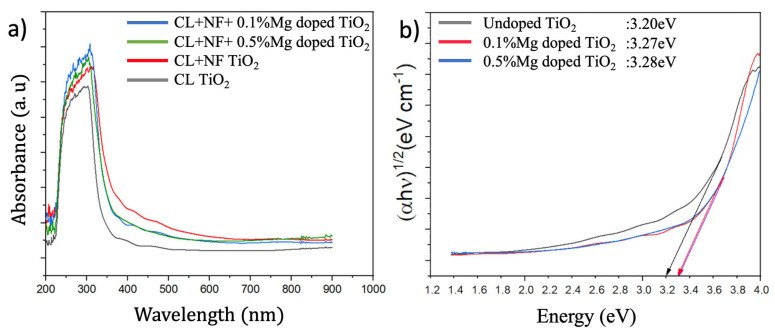
(**a**) UV-Vis absorbance spectra of TiO_2_ compact layer (CL), TiO_2_ nanofibers, and the TiO_2_ nanofibers with 0.1% and 0.5% Mg doping. (**b**) Band gap analyzes of the non-doped TiO_2_, 0.1% Mg-doped TiO_2_, and 0.5% Mg-doped TiO_2_ nanofibers.

**Figure 9 nanomaterials-13-02255-f009:**
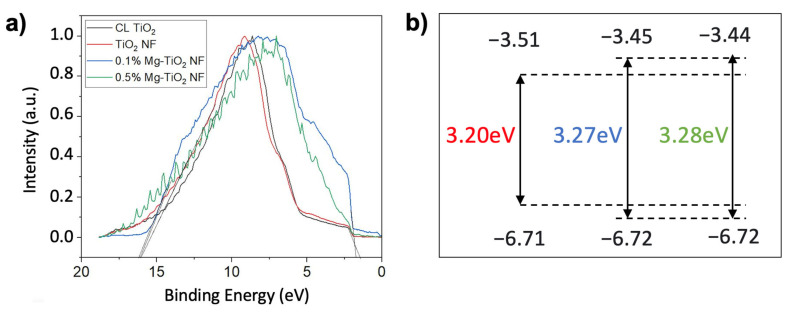
(**a**) UPS spectra of non-doped TiO_2_, 0.1% Mg-doped TiO_2_, and 0.5% Mg-doped TiO_2_ nanofibers. The left-hand side is the extended valence spectra, and the right-hand side is secondary-electron cut-off. (**b**) Energy level diagrams of non-doped TiO_2_ and Mg-doped TiO_2_ fiber ETM layers.

**Figure 10 nanomaterials-13-02255-f010:**
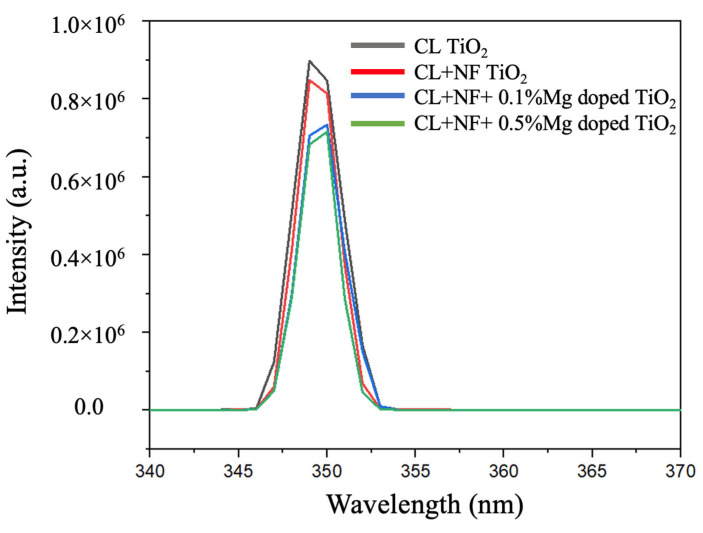
The PL emission intensity of the non-doped TiO_2_, 0.1% Mg-doped TiO_2_, and 0.5% Mg-doped TiO_2_ nanofibers.

**Table 1 nanomaterials-13-02255-t001:** Mg-doped nanofibers’ diameters before and after two-hour calcination.

Sample Name	MgCl_2_ (%)	T (°C)	D (nm)	Shrinkage (%)
A	0.1	-	249	-
B	0.1	500	80	%68
C	0.5	-	262	-
D	0.5	500	133	%50

**Table 2 nanomaterials-13-02255-t002:** (**a**) The crystallite size of non-doped TiO_2_ and MgCl_2_-doped TiO_2_ films. (**b**) The crystal size of non-doped TiO_2_ and MgCl_2_-doped TiO_2_ films with details. All films were annealed at 500 °C.

	**(a)**			
**Annealing Time (h)**	**Samples**	**2θ**	**FWHM**	**Crystal Size (nm)**
3	TiO_2_	25.25	0.88	9.22
2	TiO_2_	25.30	0.76	10.69
2	TiO_2_ + 0.1% MgCl_2_	25.25	1.23	6.61
2	TiO_2_ + 0.1% MgCl_2_	25.29	0.74	10.86
	**(b)**			
**Annealing Time & Phase**	**2θ**	**Crystal Size**	**Average Crystal Size**	
3 h & Rutile	27.37	10.67	14.11	
	36.07	13.54		
	41.24	23.10		
	54.31	9.34		
	56.59	22.09		
	69.18	5.91		
2 h & Rutile	27.42	15.55	17.80	
	36.07	17.99		
	41.25	22.17		
	54.34	15.80		
	56.65	26.28		
	69.19	9.03		
3 h & Anatase	25.25	9.22	12.23	
	37.77	19.51		
	48.03	17.39		
	55.05	2.83		
2 h & Anatase	25.30	10.69	14.48	
	36.07	16.60		
	48.08	15.81		
	62.77	14.84		
2 h & Rutile_0.1%Mg-Doped	27.24	6.65	6.82	
	54.43	3.93		
	56.91	12.45		
	69.82	4.26		
2 h & Anatase_0.1%Mg-Doped	25.25	6.61	4.06	
	37.82	3.51		
	47.99	4.37		
	62.46	1.76		
2 h & Rutile_0.5%Mg-Doped	26.09	3.38	6.25	
	35.94	10.10		
2 h & Anatase_0.5%Mg-Doped	25.29	10.86	4.00	
	37.54	0.68		
	49.66	0.45		

## Data Availability

Not applicable.

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
