# Peer review of "Investigation of Structural, Morphological, and Optical Properties of Novel Electrospun Mg-Doped TiO2 Nanofibers as an Electron Transport Material for Perovskite Solar Cells"

_nanomaterials, 2023, doi:10.3390/nano13152255_

Round 1

Reviewer 1 Report

This work reports on Mg-doped nano-fibers prepared by the electrospinning method as an ETM layer in HPSCs. They have achieved nice nanorod or nano-fibers SEM images and are characterized by XRD and other opto-physical properties. Though they have not given the device data. This work is worthful. This work can be considered after addressing the following comments.

Comments

1.       Since this is about deposition methods and oxide semiconductors. It is important to include some discussion on the oxide transport layer beside TiO2. For example; NiOx (https://doi.org/10.1016/j.tsf.2022.139486) , CuO2 (https://doi.org/10.1002/smll.201501330), SnO2 etc

2.       Please put HOMO and LUMO level in Figure 1b.

3.       If possible, please include Mg-elemental mapping in the SEM image. It is quite interesting.

4.       I am curious about its cross-sectional image. Please include it and add discussion.

5.       It is strongly recommended to include the XPS spectra of Mg in Figure 6. Please provide a reliable explanation about XPS fitting and UPS analysis.

Author Response

Thank you all for your insightful and constructive comments. We have benefited from these comments and substantially improved the quality of the manuscript. Note that, in common with these general comments, all our responses are written in red, whereas the reviewer comments are in bold.

Reviewer 2 Report

Attached, I have added two versions of the edited manuscript (I had two working copies on different computers, so I merged them). In the same file there is a paper that should also be cited.

I guess from a science perspective, I am not sure the conclusions of Mg-doping is accurate. The XRD were not well resolved, and there is an unexplained peak in the non-doped spectra (see notes on the PDF). How do you know that these extra peaks are not from graphitic formation upon calcination of PVP or merely MgO nanoparticles on the surface.

I would be happy to review this work again in greater detail if the authors can address the point above and fix the grammar in the document.

Author Response

(The authors gave the same response as above.)

Round 2

Reviewer 1 Report

The revised manuscript has not improved in scientific content.

"""

Reviewer comments>>>If possible, please include Mg-elemental mapping in the SEM image. It is quite interesting.

 Response>we don’t have any samples. It may take time (approximately 1 week) to prepare new samples. 

Review comment >>>> I am curious about its cross-sectional image. Please include it and add discussion.

Response>>Thank you for this comment. However, we had only 10 days to provide feedbacks.. It may take time (approximately 1 week) to prepare samples for cross section analysis with another student.....

**

The reply so no non-professional. There cannot be any excuse for the quality. 

I am so sorry I am unable to accept manuscript at this stage.

Author Response

Thank you all for your insightful and constructive comments. We have benefited from these comments and substantially improved the quality of the manuscript. The required revision has been made in the revised manuscript. Our point-by-point response to the your comments is also submitted.

Reviewer 2 Report

The edits that I suggested seem to be made, now I think this can be accepted with minor revision (grammar/english/formatting)

Just some minor remaining grammar, but that should be caught in the Galley proofs

Author Response

(The authors gave the same response as above.)

Round 3

Reviewer 1 Report

Accept